# COVID-19 and *Vasa vasorum*: New Atherogenic Factor? A Case Report and Autopsy Findings

**DOI:** 10.3390/diagnostics13061097

**Published:** 2023-03-14

**Authors:** Julia A. Macarova, Sofia A. Malakhova, Tatiana A. Novitskaya, Valeria A. Shapkina, Leonid P. Churilov

**Affiliations:** 1Laboratory of the Microangiopathic Mechanisms of Atherogenesis, Saint Petersburg State University, 199034 Saint-Petersburg, Russia; 2Department of Experimental Tuberculosis, St. Petersburg State Research Institute of Phthisiopulmonology, 191036 Saint-Petersburg, Russia

**Keywords:** COVID-19, SARS-CoV-2, atherosclerosis

## Abstract

Patients with COVID-19 demonstrate higher rates of cardiovascular complications, including thromboses and thromboembolism. One may suppose that the action of SARS-CoV-2 transforms stable atherosclerotic plaques into unstable status. Cardiovascular complications in COVID-19 may be caused by progressive viral alteration of the blood vessels, including *Vasa vasorum*. A lethal case of ischemic brain disease caused by cerebral atherosclerosis and exacerbated by a stroke during COVID-19 infection is briefly described. The results of the autopsy showed perivascular lymphocytic infiltration and signs of *Vasa vasorum* vasculitis with thrombi of adventitial microvasculature. The data discussed in the article are interpreted in the context of the concept giving the important role in atherogenesis to *Vasa vasorum*.

## 1. Introduction

The novel Coronavirus infection (COVID-19), caused by SARS-CoV-2, was characterized as a pandemic by WHO in March 2020 [1].

There is a growing number of publications dedicated to the interactions between SARS-CoV-2 and various human tissues and organs. COVID-19, unlike other respiratory viruses, often causes cardiovascular complications, i.e., myocarditis, arrhythmia, cardiac failure, thromboembolism, disseminated intravascular coagulation, etc. [2].

Patients with COVID-19 may develop severe vascular damage, which may persist, to some extent, in long COVID. This may explain a higher frequency of vascular complications during and after COVID-19 [3].

Different authors have described several mechanisms for the development of microcirculation disorders, including endothelial dysfunction, changes in blood rheology and formation of microtrombi [4,5,6].

Microcirculation disorders affect small vessels, including vasa vasorum, which provide blood support for adventitia and media of vessels [7]. Arterial vasa vasorum are functional end arteries [8], meaning that they do not form anastomoses with collateral vessels. There are two kinds of vasa vasorum described in contemporary publications: vasa vasorum interna, which take their origin in the lumen of the artery, and vasa vasorum externa, which are situated near and inwards the tunica adventitia of the arteries [9,10]. Disorder of perfusion or thrombosis of these small vessels may induce hypoxia, which affects the function and structure of the arterial wall.

There is growing evidence that patients who survived COVID-19 (as well as some other severe endotheliotropic viral infections) may develop vasa vasorum damage [8,11,12]. Boyle and Haverich have shown that large vessel damage in COVID-19 infection both in adults and children is possibly associated with vasa vasorum dysfunction. SARS-CoV-2-induced microthromboses in vasa vasorum may cause the hypoxia of vascular adventitia [8].

Patients often develop arterial thromboembolic complications during or after COVID-19 infection [13]. These complications may cause acute myocardial infarction or acute ischemic stroke [14,15]. Hence, one may suppose that SARS-CoV-2 is able to change the status of atherosclerotic plaques [16].

Atherosclerosis is definitely the most frequent cause of ischemic heart disease, cerebrovascular disease and diseases of peripheral arteries. The most severe consequences of atherosclerosis, such as ischemic stroke and myocardial infarction, are caused by plaque formation and plaque-induced thromboembolism.

Multiple investigations dedicated to atherosclerosis support the suggestion that atherosclerosis is a reaction to vascular injury and inflammation. Endothelial dysfunction is supposed to be a factor of atherosclerosis progression. A. Haverich supposes that the first pathophysiologic mechanism inducing inflammation of vessels may be vasa vasorum injury or thrombosis, which spreads from arterial adventitia to intima [8,9]. Ischemia, which is induced by vasa vasorum injury, leads to the formation of atherosclerotic in the zones of the artery, which are supplied with blood by that vasa vasorum. Fatal atherosclerotic complications are associated also with epicardial adipose tissue around coronary arteries [17].

The recent data about the worsening of the course of atherosclerosis in patients with COVID-19 infection, the presence of inflammation of arterial wall both in COVID-19 and atherosclerosis, the reports of higher frequencies of cardiovascular complications in COVID-19 and post-COVID-19 disorders, and the importance of *Vasa vasorum* in atherogenesis all call for the investigation of pathomorphology of vessel walls in these diseases [7,18].

## 2. Case Presentation

The patient, a 69-year-old woman with arterial hypertension, survived COVID-19, of moderate severity, in April of 2022. In May 2022, she developed acute ischemic stroke and, despite being given medical aid, died of progressive brain edema.

During the post-mortem examination of aorta and large arteries, multiple atherosclerotic plaques were found. Some of them were yellow, merging with each other, forming larger confluent plaques. Additionally, thick, firm, yellowish-white plaques were found rising over the surface of intima, making the wall of aorta appear lumpy. Some of the plaques were bearing intraplaque hemorrhages, foci of necrosis, or thrombotic masses on their surface.

Samples of aorta were taken to perform pathohistological examination, fixed in 10% neutral buffered formalin and embedded in paraffin. Standard H&E, Alcian blue and van Gieson stainings were performed.

Microscopic examinations of the aorta found atherosclerotic plaques with various morphological characteristics: plaques with well-defined fibrous caps and small atheromatous nuclei; plaques with thin fibrous cap and relatively vast lipid deposits (Figure 1a) and plaques with tears and fissures of fibrous caps (Figure 1b).

Neovascularization was found in the zones of aortic wall, corresponding to the plaques.

In the zones corresponding to unstable plaques, moderate thickening of *Vasa vasorum* walls was found due to focal deposition of glycosaminoglycans (Figure 2a,b).

There were found degenerative changes of *Vasa vasorum* endothelium (Figure 3), as well as stasis of erythrocytes and red thrombi in *Vasa vasorum* lumina (Figure 4a,b).

In aortic adventitia, mild to moderate lymphocytic infiltrate was present in the loci corresponding to focal oedema of *Vasa vasorum* (Figure 5).

Additionally, in the zones corresponding to thrombotic *Vasa vasorum* intima of the aorta demonstrated signs of endotheliitis, including moderate nuclear polymorphism and swelling of cytoplasm of endothelial cells (Figure 6).

## 3. Discussion

Thus, clinical and pathomorphological characteristics of the case are consistent with the hypothesis that *Vasa vasorum* alteration may be a common link in the pathogenesis of COVID-19 and atherosclerosis, regarding arterial damage progression. Severe cardiovascular complications of COVID-19 may be associated with progressive viral alteration of endothelium, including that of *Vasa vasorum*.

*Vasa vasorum* supply the walls of large vessels with blood.

The function of *Vasa vasorum*, similar to other vessels, is controlled by multiple substances, including vasoactive mediators and pro- and anti-inflammatory autacoids. A decrease in blood flow in adventitial *Vasa vasorum* may cause hypoxia of the arterial wall and the formation of arterial aneurisms [19,20].

*Vasa vasorum* inflammation may be the cause of the development of cardiovascular dysfunction and the formation of aneurisms in Kawasaki disease and other diseases. Histologic examination of Kawasaki disease shows perivasculitis, chronic inflammation along the coronary arteries and inflammation of their adventitia [21].

Multiple cases of Kawasaki-like disease were described in patients with acute COVID-19 or post-COVID-19 syndrome. The disease was marked by the development of arteriitis and thromboses and appeared to be one of the typical complications of COVID-19 [8].

The studies of the experience gained during the COVID-19 pandemic show that COVID-19, acknowledged as a “virus of autoimmunity” [22], as well as COVID-19-induced systemic hypercytokinaemia, naturally damages vascular endothelium, provoking thrombotic complications and also Kawasaki-like arteriites [20], interact with some of the proteins that play a role in atherosclerosis development, and, possibly, may accelerate atherogenesis and development of complications of atherosclerosis [23]. We may suppose that *Vasa vasorum* are the key point that undergoes damage, as was shown in the case study of the aneurism of the aorta during COVID-19 infection [7].

The impact of SARS-CoV-2 on the blood vessels is mediated by angiotensin-converting enzyme 2 (ACE2) expressed on the surface of the cells, interacting with the virus [24,25]. ACE2 is plentifully expressed not only on alveolocytes and small intestine enterocytes, but also on endotheliocytes, both arterial and venous [26]. Hyperexpression of ACE2 during the COVID-19 infection may cause endothelial dysfunction and alteration of endotheliocytes [27], which is associated with cardiac failure, acute coronary syndrome and other multiorganic thrombovascular complications [10].

Alteration of the cardiovascular system by SARS-CoV2 may be caused not only by direct primary action of the virus itself, but also may result from secondary alteration, induced by excessive activity of pro-inflammatory mediators. There is a certain similarity found between pro-inflammatory changes, characteristic of COVID-19 and those typical for atherosclerosis [3].

SARS-CoV-2 binds the membrane ACE2 of the endothelial cells. They express receptors of interleukin-6 [28] and interleukin-2 [29] (IL-6R and IL-2R). Direct and indirect interaction of multiple cytokines, e.g., IL-1beta, IL-6, IL-2R and TNF [16], which are abundant in the blood of COVID-19 patients, with endotheliocytes induces further alteration of endothelium. Continuous endothelial damage results in additional secretion of pro-inflammatory autacoids, enhancing the systemic action of pro-inflammatory mediators which may cause hemodynamic shock [3,27].

The action of SARS-CoV on aorta and other large vessels includes not only direct interactions between the virus and endothelium and the action of inflammatory mediators. Recent data support the hypothesis that SARS-CoV2 may damage not only endothelium of the large vessels, but also their adventitia, inducing *Vasa vasorum* inflammation. That inflammation deteriorates the blood supply of the arterial wall and enhances atherogenesis [30].

A paper published in 2021 described a clinical case of occlusion of lower limb arteries of a 77-year-old woman with COVID-19 pneumonia [31]. The patient was admitted to hospital with fever and trophic disorders of left lower limb. A deep venous thrombosis of the left soleal vein, a steno-obstruction of the left superficial femoral artery and occlusion of the ipsilateral posterior tibial artery were diagnosed during ultrasound investigation. The patient received therapy with anticoagulants. Endovascular surgery was performed. In spite of that, the patient’s signs of limb ischemia were increasing, so the leg was amputated above the knee.

Histologic examination of the arteria revealed significant stenosis. Arterial walls did not show any signs of atherosclerosis. The intima of the vessels was moderately thickened due to excessive deposition of mucopolysaccharides. Pools of mucin were also found. The endothelial lining (CD34-positive) was intact. The inner elastic laminae were intact as well. Tunica media of the arteries displayed fibrosis. No signs of inflammation of tunica media were found. There were fresh or recanalized arterial thrombi in some sections of the vessels.

Similarly, in our case, lymphocytic infiltrate, mild-to-moderate, was found in the adventitia. The adventitial *Vasa vasorum* in the zones of infiltration were demonstrating focal oedema and endotheliitis [31].

In the same year, another clinical case was published that supports the hypothesis of COVID-19 impact on atherogenesis [7]. A 67-year-old man died of respiratory failure caused by COVID-19. A post-mortem examination revealed multiple atherosclerotic plaques. Histological investigation of the samples of aorta showed lymphocytic infiltrate around *Vasa vasorum* and multiple microtromboses of their lumina, similar to the case we describe in this article.

Atherosclerosis is a chronic inflammatory disease, which develops in elastic and muscular-elastic arteries, whilst COVID-19 is regarded as a viral infection, which first affects the lungs, and later also the cardiovascular system. Hypoxemia, oxidative stress and elevated pulmonary pressure, which is caused by pulmonary thrombosis, induce cardiac dysfunction and heart failure [32]. Severe tissue hypoxia induces the production of pro-inflammatory mediators by necrobiotic cells, which results in failure of inflammatory focal barriers and systemic correlates of inflammatory process. In extreme degrees of hyperautacoidemia and excessive systemic action of inflammatory mediators, “cytokine storm” follows, resulting in distributive shock, which aggravates hypoxia [27,32].

Atherosclerotic plaques are the main pathomorphological elements of atherosclerosis. In clinical terms, they may be either stable or unstable.

The state of atherosclerotic plaque, according to forensic and post-mortem studies, significantly affects the risk of sudden cardiac death, although thrombogenic and vasospastic events in post-COVID patients also depend on many external factors [17,33].

In a pathophysiologic sense, it means to be either in a status of an inflammatory focus effectively isolated by the functional and structural barriers, or, alternatively, in a status of inflammatory focus with barriers failed and pro-inflammatory signals emitted to circulation and neighboring areas. The main attributes of stable plaques are thick, firm, collagen-rich fibrous caps, smaller quantities of active macrophages and extracellular lipids. Unstable plaques, by contrast, have thin fibrous caps, large lipid nuclei and many active macrophages and demonstrate signs of active inflammation.

The development of atherosclerotic plaques is a complicated process, including multiple cascades, where both pro- and anti-inflammatory mediators secreted by endothelial cells, leukocytes, platelets, foam cells and mast cells take part [13,34]. Insatiable active plaques disturb the blood flow not only biomechanically, due to their geometry, but also for their being a source of thrombogenic and vasoconstrictive inflammatory autacoids of lipid and peptide structure secreted by plaque cells [35,36]. Hence, both in COVID-19 and in atherosclerosis, one of the essential matters is the integrity of barriers delimiting the spheres of action for local autacoids of para-, juxta- and autocrine action (like cytokines), and preventing them from the excessive systemic spread and jeopardizing of neuroendocrine regulation supporting the whole-body vital functions [27,30].

Endothelial injury, dyslipidemias and shear stress followed by flow-mediated changes belong to the main causes of plaque formation and progression of atherosclerosis [37].

At the moment, there are two main concepts of atherogenesis: the common “inside-out” concept and the opposite “outside-in” one [38,39].

According to the “inside-out” concept, atherogenesis includes endothelial activation, the release of pro-inflammatory cytokines and chemokines, and activation of inflammatory cascades. The atherogenic lipoproteins (LP) are accumulated in the intima, oxidized and modified. The cellular adhesion molecules expressed on activated endothelial cells bind circulating monocytes. The monocytes migrate into intima and then transform into macrophages, expressing LP-binding receptors. After the accumulation of the excess of LP, the macrophages transform into foam cells. T-lymphocytes and smooth muscle cells (SMC) from the arterial media also migrate into intima, and SMC take part in foam cell formation.

According to the “outside-in” concept of atherosclerosis, the inflammation begins in adventitial layer of the vessel and then spreads on the media and intima [9,31,38]. In this concept, *Vasa vasorum* play an important role in the formation of atherosclerotic changes and in the provocation of the complications of atherosclerosis [8,9,10].

It is well-known that microvasculature is being damaged in many autoimmune and viral diseases, particularly in Kawasaki disease, when lesions involve many arteries similarly involved in atherosclerosis. Interestingly, Kawasaki-like syndrome was described as a complication of COVID-19 in children. Perhaps *Vasa vasorum* vasculitis may be the common thread between COVID-19, autoimmune vasculitis and atherosclerosis [8,40].

The presence of microthrombi in *Vasa vasorum* indicates that vascular injury associated with COVID-19 develops in microvasculature first as it was demonstrated earlier by Boyle et al. [8].

On the other hand, damage of *Vasa vasorum* may possibly shift the stable atherosclerotic plaques into unstable conditions, increasing the risk of development of atherosclerotic complications, which obviously happened in the described case of post-COVID ischemic stroke.

The similarly central role of *Vasa vasorum* and their viral/immunopathological damage was earlier noted not only in atherosclerosis, but also in Takayasu arteritis, Buerger’s disease, temporal arteritis, vascular form of Behcet disease and inflammatory abdominal aortic aneurysm [29], as well as in Churg-Strauss syndrome [41]. The authors cited above related these immunopathological disorders to atherosclerosis, and F. Numano [42] even coined the non-conventional term: “vasa-vasoritis”.

## 4. Conclusions

Morphological characteristics of this case are consistent with the suggestion that *Vasa vasorum* damage may be a common link in the pathophysiology of COVID-19 infection and atherosclerosis, concerning the progression of arterial lesions.

The action of SARS-CoV-2 on large vessels may appear to be more complicated than it was supposed to be in recent times. The virus may possibly induce an inflammatory process and dysfunction of endothelial cells not only in the intima of the arteries, but also disfunction of *Vasa vasorum* in the adventitia.

This process may lead to a disruption of blood support of the vascular wall and thus accelerate atherogenesis and provoke its complications.

The number of persons who have survived COVID-19 rapidly increases all over the world, and a considerable part of this cohort, especially elderly patients, presumably already suffer from atherosclerosis [43]. Bearing in mind the data described above, one may expect that new Coronavirus infections will soon display their atherogenic and complication-provoking potential. Perhaps COVID-19 will be added to the list of atherosclerosis risk factors.

## Figures and Tables

**Figure 1 diagnostics-13-01097-f001:**
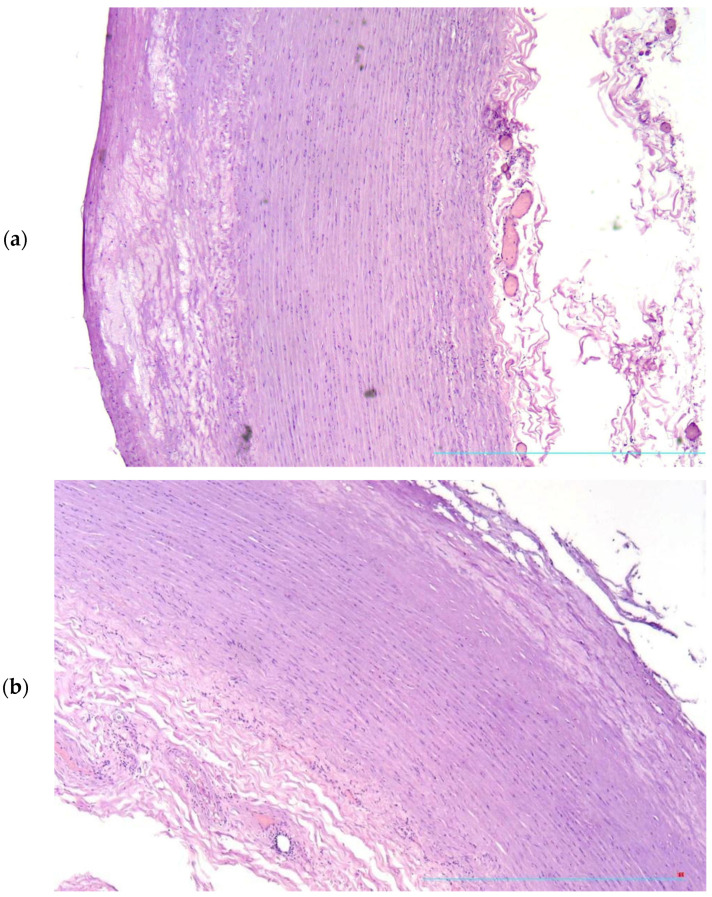
Thinning (**a**) and ruptures (**b**) of fibrous cap of the atherosclerotic plaque. H&E, ×40.

**Figure 2 diagnostics-13-01097-f002:**
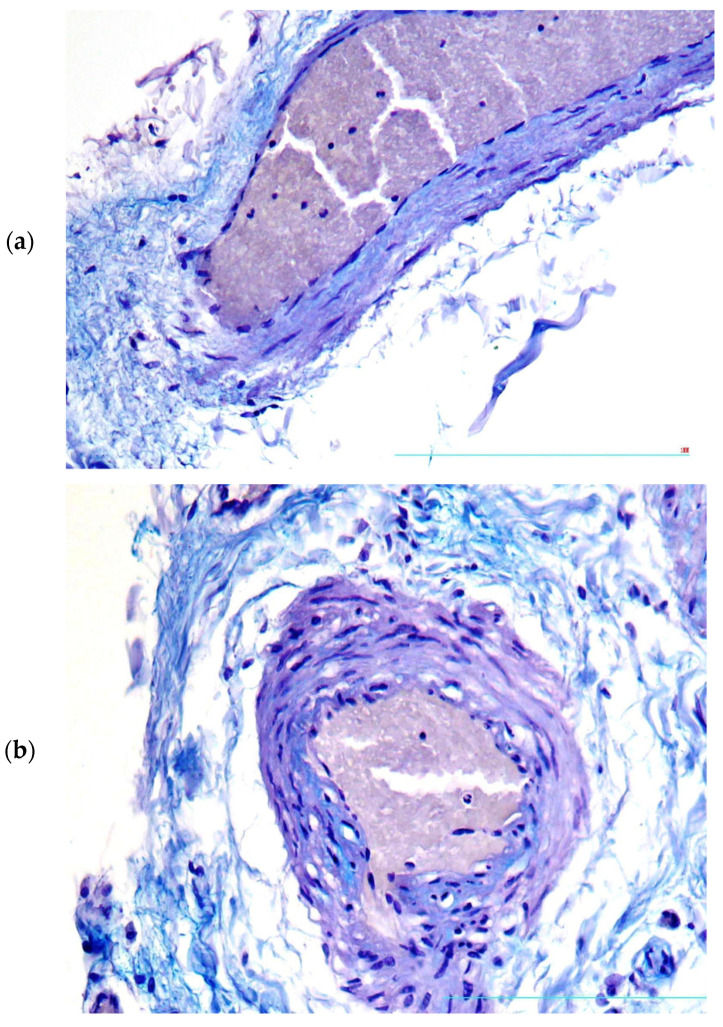
(**a**,**b**) Focal deposition of glycosaminoglycans in *Vasa vasorum* walls, Alcian blue, ×200.

**Figure 3 diagnostics-13-01097-f003:**
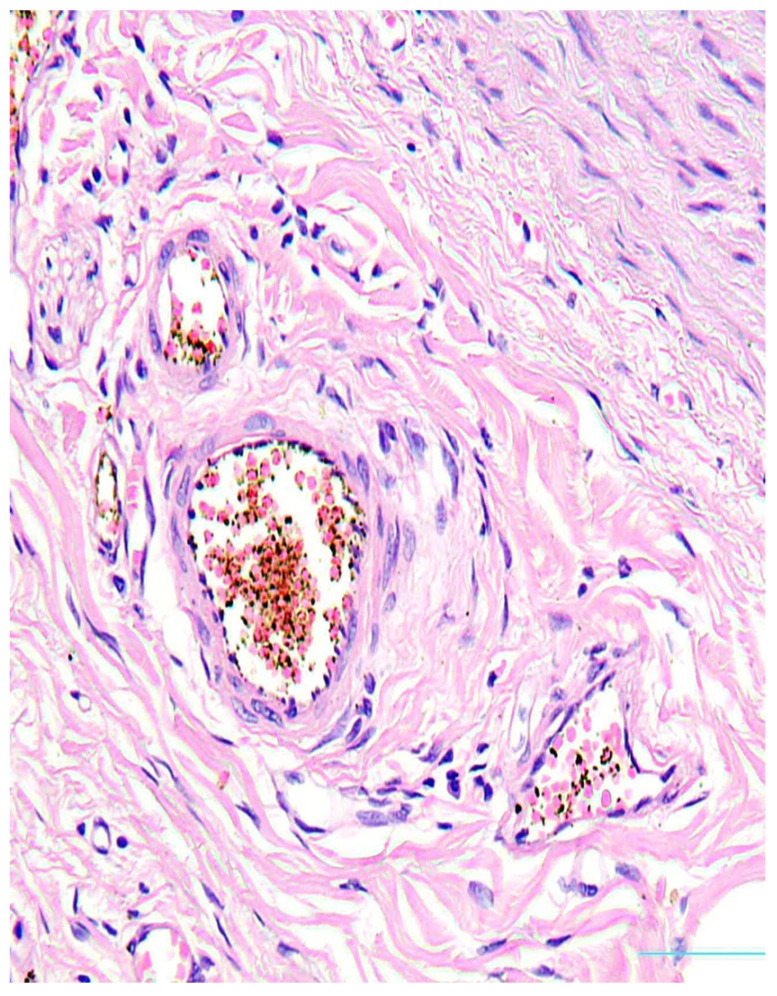
Degeneration of *Vasa vasorum* endotheliocytes. H&E, ×200.

**Figure 4 diagnostics-13-01097-f004:**
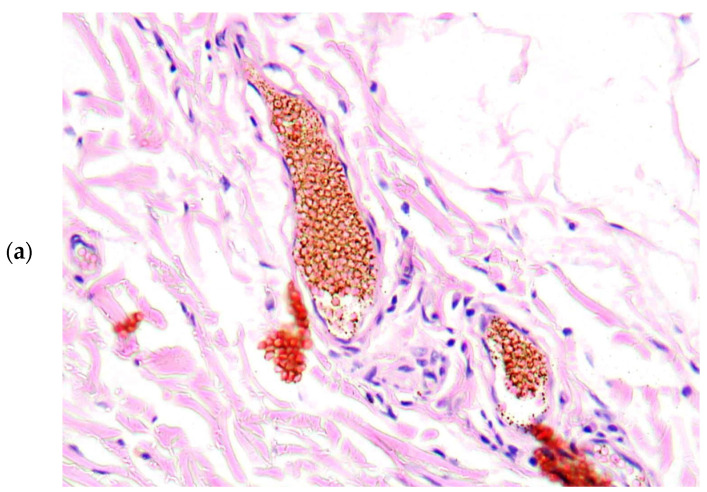
Stasis of erythrocytes (**a**), H&E, ×100 and thrombi (**b**) in *Vasa vasorum* lumina, H&E, ×200.

**Figure 5 diagnostics-13-01097-f005:**
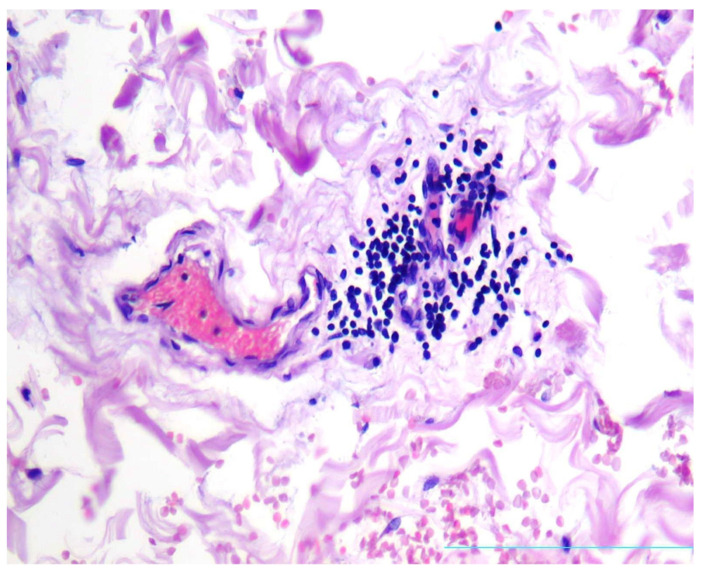
Mild perivascular lymphocytic infiltration, H&E, ×100.

**Figure 6 diagnostics-13-01097-f006:**
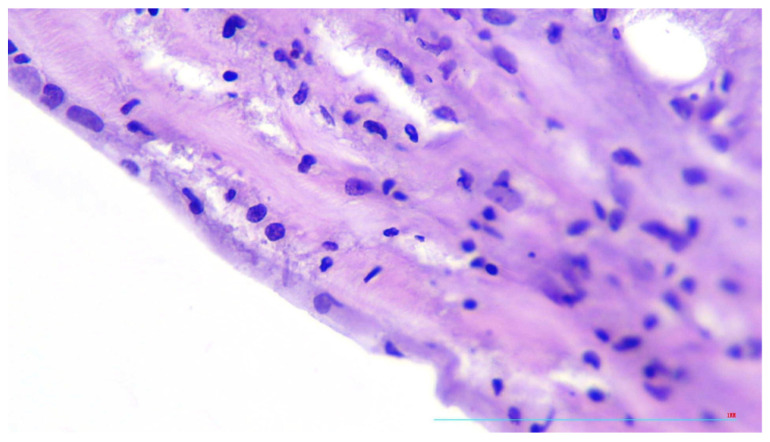
Focus of endotheliitis of aorta, H&E, ×200.

## Data Availability

Not applicable.

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
