# Peer review of "COVID-19 and Vasa vasorum: New Atherogenic Factor? A Case Report and Autopsy Findings"

_diagnostics, 2023, doi:10.3390/diagnostics13061097_

Round 1

Reviewer 1 Report

This is an interesting paper that fits the Journal scope. The authors underlying the vasa vasorum role in atherogenesis, from a forensic perspective

Is well known that COVID-19 infection increase the risk of thromboembolic events, especially when an atherosclerotic process is present.

Major comments:

1.     I suggest to the authors to reduce the Introduction section and redistributed the information to Discussion section.

2.     I suggest authors to improve the case presentation with comorbidities of patients, treatment of the AIS (medication, thrombolysis, thrombectomy etc)

3.     There is recent publication that will improve the article quality. I suggest authors discuses in the Discussion section the results from the following article:

- https://doi.org/10.3390/diagnostics13010142

- https://doi.org/10.3390/diagnostics12112757

Minor comments:

1.     References 11 and 12 are duplicated. Please remove one and rearrange the references.

Author Response

Thank you for your comments.

Point1.  I suggest to the authors to reduce the Introduction section and redistributed the information to Discussion section.

Response 1.We have reduced the Introduction section and transfered a part of the Introduction to Discussion.

Point 2.  I suggest authors to improve the case presentation with comorbidities of patients, treatment of the AIS (medication, thrombolysis, thrombectomy etc).

Response 2.The patient had only medication treatment. There was no intravascular intervention or thrombolysis.

Point 3. There is recent publication that will improve the article quality. I suggest authors discuses in the Discussion section the results from the following article:

Response 3. The articles were quite helpful, thank you

We have managed with duplicated references.

Reviewer 2 Report

This is a case report with an interesting suggestion. Another possibility is the complications of the vaccines and that we have had several cases of aortic thrombosis in young and elderly patients and, when arteriosclerosis is present, the presence of a thrombus on the plaque.

Author Response

Thank you for your comments. Here is our updated article 

Reviewer 3 Report

1.      T     1.                   The title of the article “COVID-19 and Vasa Vasorum: New Atherogenic Factor?” is not quite right in my opinion.  What exactly is the new factor?

2.                   The introduction is the main part of the article. It contains information that is not directly related to the content of the paper. The introduction could be shortened.

3.                   In the introduction the authors write “ SARS-CoV-2 binds the membrane ACE2 of the endothelial cells. They express receptors of interleukin-6 and interleukin-2 (IL-6R and IL-2R). Binding of IL-2 and IL-6 abundant in blood of COVID-19 patients with endotheliocytes induces further alteration of endothelium. Continuous endothelial damage results in additional secretion of pro-inflammatory autacoids, enhancing systemic action of pro-inflammatory mediators which may cause hemodynamic shock [21]”. First, I have doubts that IL-2 receptors are expressed by endothelium. There is no information about this in reference (21) given by the authors. Secondly, much more cytokines are involved in the development of endothelial dysfunction in viral infections [see for example Am J Med Sci 2022;363(4):281-287].

4.                   The authors write: Standard H&E, alcian blue and Van Gieson stains were performed. Below are photographs of slices stained only with H&E.

5.                   Figures showing a section of the entire aorta with indication of the locus, which was then more thoroughly analyzed, should be provided.  For example, see Faa et al, Eur Rev Med Pharmacol Sci . 2021 Oct;25(20):6439-6442. Were there atherosclerotic plaques at the level of the analyzed sites; if so, what was their morphology?

6.                   The authors write “During microscopic examination of aorta and large arteries there were found a moderate thickening of vasa vasorum walls due to excessive deposition of glycosaminoglycans, as well as degenerative changes of endothelium (Figure 1)”. How was this determined? (Slices stained with alcian blue and Van Gieson are not shown).

7.                   Figure 2 demonstrates "Stasis of the erythrocytes (a) and thrombi in vasa vasorum”. The accumulation of erythrocytes in a vessel on a slice of tissue can be a phenomenon not related to a pathological process in a particular site. As a rule, in erythrostasis, erythrocyte clusters form coin columns, loops, or large conglomerates. The structures indicated as "thrombus" may also result from hemolysis. The authors should provide clarification on this point.

8.                   Another question arises, were the slices showing vasa vasorum obtained from the same tissue specimen?

9.                   Figure 3 shows “Subtle perivascular lymphocytic infiltrate around vasa vasorum of the aorta”. I can't distinguish lymphocyte nuclei from monocyte nuclei in this picture.

10.                Figure 4. Specific signs of endotheliitis should be clarified.

11.                In general, based on the data presented in the article, one cannot conclude that the atherosclerotic plaque destabilization and stroke development in this patient is a result of a pathological process in the vasa vasorum, provoked by SARS-Cov2 infection.  

Author Response

Point 1.The title of the article “COVID-19 and Vasa Vasorum: New Atherogenic Factor?” is not quite right in my opinion.  What exactly is the new factor?

In the title of the article we suggested that COVID-19, inducing vasa vasorum vasculitis, may deteriorate the course of atherosclerosis in some patients.

Point 2. The introduction is the main part of the article. It contains information that is not directly related to the content of the paper. The introduction could be shortened.

Response 2. We have shortened the Introduction and transmitted a part of it to the Discussion.

Point 3. In the introduction the authors write “ SARS-CoV-2 binds the membrane ACE2 of the endothelial cells. They express receptors of interleukin-6 and interleukin-2 (IL-6R and IL-2R). Binding of IL-2 and IL-6 abundant in blood of COVID-19 patients with endotheliocytes induces further alteration of endothelium. Continuous endothelial damage results in additional secretion of pro-inflammatory autacoids, enhancing systemic action of pro-inflammatory mediators which may cause hemodynamic shock [21]”. First, I have doubts that IL-2 receptors are expressed by endothelium. There is no information about this in reference (21) given by the authors. Secondly, much more cytokines are involved in the development of endothelial dysfunction in viral infections [see for example Am J Med Sci 2022;363(4):281-287].

Response 3. Thank you for your comment. It is our mistake.

The situation with IL-2R and endotheliocytes appears to be quite strange.

There is some date that IL-2R is expressed on endotheliocytes. Improved IL-2 immunotherapy by selective stimulation of IL-2 receptors on lymphocytes and endothelial cells - PMC (nih.gov)

Also binding of IL-2 with endothelial cells elevates vascular permeability. Interleukin 2 Activates Brain Microvascular Endothelial Cells Resulting in Destabilization of Adherens Junctions - PMC (nih.gov)

It is mentioned in the table 1 of the article Inflammatory Mechanisms in COVID-19 and Atherosclerosis: Current Pharmaceutical Perspectives - PMC (nih.gov) (reference 21), that IL-2 and its receptor are elevated in COVID and correlate with its severity. But the article the authors cite  CT Imaging of the 2019 Novel Coronavirus (2019-nCoV) Pneumonia - PMC (nih.gov) doesn’t even mention interleukins.

Point 4.         The authors write: Standard H&E, alcian blue and Van Gieson stains were performed. Below are photographs of slices stained only with H&E.

Response 4. We have added the photos of the slides stained with alcian blue. Van Gieson staining appeared to be not very demonstrative, so we didn’t make any photos. We may just remove the mentioning of it from the article.

Point 5.    Figures showing a section of the entire aorta with indication of the locus, which was then more thoroughly analyzed, should be provided.  For example, see Faa et al, Eur Rev Med Pharmacol Sci . 2021 Oct;25(20):6439-6442. Were there atherosclerotic plaques at the level of the analyzed sites; if so, what was their morphology?

Response 5. We have added a pair of photos of a full-thickness section slides of the aorta. At the level which was analyzed there was a plaque with a thinned fibrous cap and a vast, lipid-rich nucleus (Figure 1).

Point 6.  The authors write “During microscopic examination of aorta and large arteries there were found a moderate thickening of vasa vasorum walls due to excessive deposition of glycosaminoglycans, as well as degenerative changes of endothelium (Figure 1)”. How was this determined? (Slices stained with alcian blue and Van Gieson are not shown).

Response 6. We have added aclian blue slides (Figure 2).

Point 7.                   Figure 2 demonstrates "Stasis of the erythrocytes (a) and thrombi in vasa vasorum”. The accumulation of erythrocytes in a vessel on a slice of tissue can be a phenomenon not related to a pathological process in a particular site. As a rule, in erythrostasis, erythrocyte clusters form coin columns, loops, or large conglomerates. The structures indicated as "thrombus" may also result from hemolysis. The authors should provide clarification on this point.

Response 7. Figure 2a demonstrates quite a big conglomerate if intact erythrocytes in the lumen of the vessel, which is characteristic of erythrostasis. The accumulation of erythrocytes, shown on Figure 2b, is attached to the wall of the vessel, letting us suppose it to be a thrombus.

Point 8.                   Another question arises, were the slices showing vasa vasorum obtained from the same tissue specimen?

Response 8. Yes

Point 9.                   Figure 3 shows “Subtle perivascular lymphocytic infiltrate around vasa vasorum of the aorta”. I can't distinguish lymphocyte nuclei from monocyte nuclei in this picture.

Response 9. Monocytes are relatively large, oval shaped cells with abundant cytoplasm. Lymphocytes are relatively small, round, their cytoplasm is scant, so that he nucleus occupies the majority of the cell, as it is shown on the Figure 3.

Point 10.   Figure 4. Specific signs of endotheliitis should be clarified.

Response 10. The figure demonstrates mild polymorphism of the nuclei and oedema of cytoplasm of endotheliocytes.

Point 11.                In general, based on the data presented in the article, one cannot conclude that the atherosclerotic plaque destabilization and stroke development in this patient is a result of a pathological process in the vasa vasorum, provoked by SARS-Cov2 infection.  

Response 11. In the article we don’t discuss the patient’s stroke. It is about the findings in the aorta. The point is that the patient without diagnosed vasculitis developed vasculitis of aortal vasa vasorum. In the zones of vasculitis atherosclerotic plaques have thinned fibrous caps with tears. The patient survived COVID, for which it is characteristic to cause vasculopathies, and also COVID-19 has common traits with atherosclerosis. Also the patient had no other causes to develop vasculitis of vasa vasorum, so that we may suppose that it was associated with COVID-19.

Round 2

Reviewer 1 Report

no further comments

Author Response

Thank you

Reviewer 3 Report

The authors have responded to the comments. The article can be published after minimal revision.

1. Introduction. The statement about the presence of "the similarities between pathogenetic mechanisms of atherosclerosis and COVID-19" is unclear, should be rephrased. 

2. The text should be checked for typos. For example, there is an error in the word "lymphocytic" in the caption of Fig. 5.

Author Response

Point 1. Introduction. The statement about the presence of "the similarities between pathogenetic mechanisms of atherosclerosis and COVID-19" is unclear, should be rephrased. 

Response 1. We have paraphrased it. Now:" The recent data about worsening of the course of atherosclerosis in patients with  COVID-19 infection, presence of inflammation of arterial wall both in COVID-19 and atherosclerosis"

Point 2. The text should be checked for typos. For example, there is an error in the word "lymphocytic" in the caption of Fig. 5.

Response 2. We have checked them. Now it should be correct
